# Alkaloids of *Dicranostigma franchetianum* (Papaveraceae) and Berberine Derivatives as a New Class of Antimycobacterial Agents

**DOI:** 10.3390/biom12060844

**Published:** 2022-06-17

**Authors:** Viriyanata Wijaya, Ondřej Janďourek, Jana Křoustková, Kateřina Hradiská-Breiterová, Jan Korábečný, Kateřina Sobolová, Eliška Kohelová, Anna Hošťálková, Klára Konečná, Marcela Šafratová, Rudolf Vrabec, Jiří Kuneš, Lubomír Opletal, Jakub Chlebek, Lucie Cahlíková

**Affiliations:** 1ADINACO Research Group, Department of Pharmacognosy and Pharmaceutical Botany, Faculty of Pharmacy, Charles University, Heyrovskeho 1203, 500 05 Hradec Kralove, Czech Republic; wijayav@faf.cuni.cz (V.W.); marikoj2@faf.cuni.cz (J.K.); breiterk@faf.cuni.cz (K.H.-B.); elkohelova@gmail.com (E.K.); hosta4aa@faf.cuni.cz (A.H.); safratom@faf.cuni.cz (M.Š.); vrabecr@faf.cuni.cz (R.V.); opletal@faf.cuni.cz (L.O.); chlej2aa@faf.cuni.cz (J.C.); 2Department of Biological and Medical Sciences, Faculty of Pharmacy, Charles University, Heyrovskeho 1203, 500 05 Hradec Kralove, Czech Republic; jando6aa@faf.cuni.cz (O.J.); konecna@faf.cuni.cz (K.K.); 3Biomedical Research Centre, University Hospital Hradec Kralove, Sokolska 581, 500 05 Hradec Kralove, Czech Republic; jan.korabecny@fnhk.cz (J.K.); babkovakaterina@gmail.com (K.S.); 4Department of Bioorganic and Organic Chemistry, Faculty of Pharmacy, Charles University, Heyrovskeho 1203, 500 05 Hradec Kralove, Czech Republic; kunes@faf.cuni.cz

**Keywords:** *Dicranostigma franchetianum*, Papaveraceae, isoquinoline alkaloids, berberine, antimycobacterial activity, cytotoxicity

## Abstract

Tuberculosis (TB) is a widespread infectious disease caused by *Mycobacterium tuberculosis*. The increasing incidence of multidrug-resistant (MDR) and extensively drug-resistant (XDR) strains has created a need for new antiTB agents with new chemical scaffolds to combat the disease. Thus, the key question is: how to search for new antiTB and where to look for them? One of the possibilities is to search among natural products (NPs). In order to search for new antiTB drugs, the detailed phytochemical study of the whole *Dicranostigma franchetianum* plant was performed isolating wide spectrum of isoquinoline alkaloids (IAs). The chemical structures of the isolated alkaloids were determined by a combination of MS, HRMS, 1D, and 2D NMR techniques, and by comparison with literature data. Alkaloids were screened against *Mycobacterium tuberculosis* H37Ra and four other mycobacterial strains (*M. aurum*, *M. avium*, *M. kansasii*, and *M. smegmatis*). Alkaloids **3** and **5** showed moderate antimycobacterial activity against all tested strains (MICs 15.625–31.25 µg/mL). Furthermore, ten semisynthetic berberine (**16a**–**16k**) derivatives were developed and tested for antimycobacterial activity. In general, the derivatization of berberine was connected with a significant increase in antimycobacterial activity against all tested strains (MICs 0.39–7.81 μg/mL). Two derivatives (**16e**, **16k**) were identified as compounds with micromolar MICs against *M. tuberculosis* H37Ra (MIC 2.96 and 2.78 µM). All compounds were also evaluated for their in vitro hepatotoxicity on a hepatocellular carcinoma cell line (HepG2), exerting lower cytotoxicity profile than their MIC values, thereby potentially reaching an effective concentration without revealing toxic side effects.

## 1. Introduction

*Dicranostigma* Hooker f. & Thomson is a small genus of the Papaveraceae family with eight accepted species native to the Himalayas and western China [1,2]. Plants of this genus are also known as eastern horned poppies. Although resembling the true horned poppies of *Glaucium*, they have stigmas with two lobes and fruit with only traces of “horns”. These plants have been used for a long time in folk medicine to treat various diseases in China. They are effective for the treatment of tonsillitis, hepatitis, sore throat, scrofula, bald scars, and scabies [3,4,5]. Some species of *Dicranostigma* have been investigated in several phytochemical and pharmacological studies. So far, more than 30 isoquinoline alkaloids (IAs) of different structural scaffolds have been either identified in or isolated from the genus *Dicranostigma* [1,6,7,8]. Alkaloids from *Dicranostigma* plants have been intensively investigated for their promising activities, such as anti-inflammatory, antibacterial, antiviral, and hepatoprotective effects [9,10,11].

*Dicranostigma franchetianum* (Prain) Fedde is an herbaceous bushy annual plant that is native to western China. In Europe, the herb is known as a cultivated plant in botanical gardens.

So far, 13 IAs have been isolated from this species, including berberine, protopine, chelerythrine, and sanguinarine [12,13,14,15].

Tuberculosis (TB) is a widespread infectious disease caused by *Mycobacterium tuberculosis* (*Mtb*). According to the Global Tuberculosis Report 2021, issued by the World Health Organization (WHO), the latent form of *Mtb* has infected about a quarter of the world’s population, but only a small part (5–10%) will develop this bacterial disease. Contrary to expectations, a significant drop in newly diagnosed cases is mentioned in the global TB report 2021 (5.8 million in 2020, 7.1 million in 2019). Unfortunately, reduced access to TB treatment and diagnosis due to the coronavirus (COVID-19) pandemic (in the context of lockdowns, concerns about the risks of going to the health care facilities, the stigma associated with similarities in the symptoms related to COVID-19 and TB, etc.) has resulted in an increase in deaths caused by *Mtb* (1.3 million in 2020 vs. 1.2 million in 2019). Until the COVID-19 pandemic, TB was the leading cause of death from a single infectious agent, ranking above HIV/AIDS (13th leading cause worldwide in 2019). However, predictions suggest that the number of people developing TB and dying from it could be much worse in 2021 and 2022. 

Although most TB forms are treatable and curable, the global treatment success rate for people treated for TB with first-line regimens was 86% in 2019 (77% among people living with HIV). Standard treatment of drug-susceptible TB comprises the administration of four first-line anti-TB drugs: isoniazid (INH), rifampicin (RIF), pyrazinamide (PZA), and ethambutol (EMB). However, resistance to all of them occurs and poses a serious public health threat; thus, the treatment of TB is becoming challenging due to the emergence of multidrug-resistant TB (MDR-TB) and extensively drug-resistant TB (XDR-TB) strains [16]. Usually, MDR-TB refers to the strains that are resistant to INH and RIF, while XDR-TB strains are resistant to all first-line anti-TB drugs and, additionally, to some second-line anti-TB drugs, including fluoroquinolones and at least one of three injectables drugs (i.e., kanamycin, amikacin, capreomycin) [17]. Testing TB drug resistance is vitally important for the treatment, including microbiological methods, rapid molecular tests, and sequencing technologies. Thus, in 2020, 71% of people with bacteriologically confirmed pulmonary TB were screened for resistance, revealing 158 thousand cases with a resistant form of TB. The success rate of treatment for MDR-TB was only 59% in 2018. 

According to the WHO report from August 2021 [18], there are currently 25 drugs that are being investigated for the treatment of TB in different stages of clinical trials [18,19]. Given the increasing resistance to *Mtb*, searching for new anti-TB agents with new chemical scaffolds is essential to combat the disease. Despite the rise of combinatorial and medicinal chemistry approaches, natural products (NPs) still hold a valuable position as a source of novel drugs, but fundamental research on NPs has started to seem like an endangered species [20,21,22]. On the other hand, isolated NPs are often endowed with non-drug-like properties, low potency, low selectivity, and other inadequate properties; thus, their core scaffolds often require extensive structure-activity optimization while also improving their ADME profile to become valid and potent drugs [22,23]. From this perspective, semisynthesis using a NP as a starting building block for chemical modification is well established in the drug discovery process. According to this approach, several semisynthetic derivatives of NPs (i.e., sampangine, cleistopholine, berberine, and galanthamine) have been studied for their antimycobacterial potential [24,25,26,27].

The interesting biological activities of the IAs and the absence of current phytochemical study of *D. franchetianum* encouraged us to examine this species. As a part of our ongoing research on IAs and their semisynthetic derivatives as potential drugs, this work reports on the isolation of alkaloids from the whole plant of *D. franchetianum*, preparation of berberine derivatives, and their in vitro antimycobacterial activity against *Mtb* H37Ra, and four other mycobacterial strains *(M. aurum*, *M. avium*, *M. kansasii*, and *M. smegmatis*).

## 2. Materials and Methods

### 2.1. General Experimental Procedures

All solvents were treated using standard techniques before use. All reagents and catalysts were purchased from Sigma Aldrich, Czech Republic, and used without purification. NMR spectra were recorded in either CDCl_3_ or CD_3_OD on a VNMR S500 (Varian, Palo Alto, CA, USA) spectrometer operating at 500 MHz for proton nuclei and 125.7 MHz for carbon nuclei at ambient temperature. ESI-HRMS were obtained with a Waters Synapt G2-Si hybrid mass analyzer of a quadrupole-time-of-flight (Q-TOF) type, coupled to a Waters Acquity I-Class UHPLC system (Waters, Millford, MA, USA). The EI-MS were obtained on an Agilent 7890A GC 5975 inert MSD operating in EI mode at 70 eV (Agilent Technologies, Santa Clara, CA, USA). A DB-5 column (30 m × 0.25 mm × 0.25 μm, Agilent Technologies, USA) was used with a temperature program: 100–180 °C at 15 °C/min, 1 min hold at 180 °C, and 180–300 °C at 5 °C/min and 5 min hold at 300 °C; and detection range *m*/*z* 40–600. The injector temperature was 280 °C. The flow rate of carrier gas (helium) was 0.8 mL/min. A split ratio of 1:15 was used. Fractionation of fractions **II** and **V** was performed with a BÜCHI Sepacore flash system ×10 equipped with a BÜCHI control unit C-620, BÜCHI fraction collector C-660, UV photometer C-640, and BÜCHI pump modules C-605 (BÜCHI, Flawil, Switzerland). TLC was carried out on Merck precoated silica gel 60 F254 and Merck precoated silica gel 60 RP-18 F254 plates. Compounds on the plate were observed under UV light (254 and 366 nm) and visualized by spraying with Dragendorff’s reagent.

### 2.2. Plant Material

The plant material used from *Dicranostigma franchetianum* came from a monoculture introduced to the Garden of Medicinal Plants at the Faculty of Pharmacy, Charles University in Hr. Králové in 2014 (seeds from the Centre of Medicinal Plants of Masaryk University in Brno). Botanical identification was performed by Prof. L. Opletal. A voucher specimen is deposited in the Herbarium of the Faculty of Pharmacy in Hradec Králové under number: CUFPH-16130/AL-540.

### 2.3. Extraction and Isolation of Alkaloids

Finely cut and dried aerial parts of *D. franchetianum* (11.87 kg) were minced and sequentially extracted with 95% EtOH (500 g of material boiled with 3 L of EtOH) for 30 min. The combined extracts were evaporated to the consistency of thin syrup, to which 6 L of distilled water at 80 °C was added, and the pH adjusted to 1–1.5 by the addition of 2% H_2_SO_4_. The suspension was filtered through Celite 545. The filtrate was alkalized by 10% Na_2_CO_3_ (pH 9–10) and extracted with CHCl_3_ (3 × 5 L). The organic layer was evaporated to give 120 g of dark brown syrup. This alkaloid summary extract was again dissolved in 2% H_2_SO_4_ (3 L), defatted with Et_2_O (3 × 2 L), and alkalized to pH 9–10 with 10% Na_2_CO_3_. The water layer was extracted with EtOAc (4 × 2 L) to give 47 g of concentrated alkaloidal extract in the form of brown syrup.

The extract (47 g), adsorbed on silica gel (ratio of media to sample 2:1) and loaded onto a PP pre-column (cartridge 40 × 150 mm, Büchi), was separated by flash chromatography (Sepacore^®^ Flash Chromatography System ×10, Büchi) on silica gel (silica gel 60, 40–63 µm, glass column 49 × 460 mm, Büchi) using a mobile phase containing light petroleum (solvent A), EtOAC (solvent B) and MeOH (solvent C). Separation of the extract was carried out for 360 min, starting with isocratic elution with solvents A and B (20% B, 15 min), followed by linear gradient elution (20% B→100% B, 150 min) and isocratic solvent elution (100% B, 15 min). Chromatography of the extract was continued by linear gradient elution with solvents B and C (100% B→100% C, 165 min), and the separation was completed by isocratic elution with solvent C (15 min). The separation flow rate was set at 110 mL/min, and UV detection was performed at 254 nm, 290 nm, 315 nm and 365 nm. A total of 208 fractions (190 mL) were collected and, based on analytical TLC, combined into 12 fractions (A–L). Separation of fractions into individual alkaloids was performed by preparative TLC on silica gel (silica gel 60 GF_254_). Conditions for preparative TLC, additional separation processes (e.g., crystallization) for each fraction, and isolated amount of each alkaloid were as follows:

Fraction **A** (250 mg, cHx:Et_2_NH, 97:3, 2×) gave compound **1** (140 mg).

Fraction **B** (190 mg, cHx:Et_2_NH, 95:5, 2×) gave compound **2** (77 mg).

Fraction **C** (434 mg, To:Et_2_NH, 99:1, 2×) gave compounds **4** (20 mg) and **5** (145 mg).

Fraction **D** (807 mg, To:cHx:EtOAc:MeOH, 45:30:35:1) gave compound **3** (149 mg).

Fraction **E** (2.2 g, To:cHx:Et_2_NH, 45:45:10) gave alkaloids **6** (720 mg) and **7** (452 mg).

Fraction **F** (8.9 g) was partially separated by preparative TLC (2.0 g, To:Et_2_NH, 96:4, 3×) into 2 zones **F1** (23 mg) and **F2** (1.5 g). Subsequently, the **F1** subfraction was separated by additional preparative TLC (EtOAc:MeOH:H_2_O 100:13:10, 2×) to give **8** (8 mg). Subfraction **F2** (28 mg) gave compound **9** (2 mg) after crystalization from the mixture EtOH:CHCl_3_.

Fraction **G** (890 mg, To:cHx:Et_2_NH, 6:3:1) gave subfractions **G1**–**G3**. Subfraction **G1** (115 mg) was recrystallized from the mixture EtOH:CHCl_3_ (1:1) to give compound **10** (56 mg). Additional preparative TLC of subfraction **G2** (78 mg, To:cHx:Et_2_NH, 5:4:1, 2×) gave compounds **11** (15 mg) and **12** (13 mg).

Fraction **H** (730 mg, EtOAc:MeOH:H_2_O:TFAA, 100:40:5:0.1, 2×) gave four subfractions **H1**–**H4**. Crystallization of subfraction **H1** (97 mg) yielded compound **13** (34 mg). Additional preparative TLC of subfraction **H3** (40 mg, EtOAc:MeOH:H_2_O:TFAA 100:40:5:0.1, 2×) gave compound **14** (3 mg).

Fraction **J** (184 mg, To:cHx:EtOAc:MeOH 45:30:35:1, 1×) gave three subfractions **J1**–**J3**. Subfraction **J1** (53 mg) yielded compound **15** (11 mg), after additional preparative TLC (To:Et_2_NH, 95:5,2×).

Fraction **L** (2.3 g) was recrystallized from EtOH to give 1.1 g of **16**.

Other fractions and subfractions have not been used to isolate alkaloids due to either the low amount or complex mixture of compounds.

### 2.4. Preparation of Berberine Derivatives

The same procedure as described previously has been used to afford the corresponding ethers. NMR and mass spectra of the synthesized derivatives (**16a**–**16k**) were in agreement with a previous report [28].

### 2.5. Antimycobacterial Screening

The antimycobacterial assay was performed with rapidly growing *Mycolicibacterium smegmatis* DSM 43465 (ATCC 607), *Mycolicibacterium aurum* DSM 43999 (ATCC 23366), and non-tuberculous mycobacteria, namely *Mycobacterium avium* DSM 44156 (ATCC 25291), and *Mycobacterium kansasii* DSM 44162 (ATCC 12478) from German Collection of Microorganisms and Cell Cultures (Braunschweig, Germany), and with an avirulent strain of *Mtb* H37Ra ITM-M006710 (ATCC 9431) from Belgian Co-ordinated Collections of Micro-organisms (Antwerp, Belgium). The technique used for activity determination was the microdilution broth panel method using 96-well microtitration plates. The culture medium was Middlebrook 7H9 broth (Sigma-Aldrich, Steinheim, Germany) enriched with 0.4% glycerol (Sigma-Aldrich, Steinheim, Germany) and 10% Middlebrook OADC growth supplement (Himedia, Mumbai, India) [29,30].

The mycobacterial strains were cultured on Middlebrook 7H9 agar and suspensions were prepared in Middlebrook 7H9 broth. The final density was adjusted to 1.0 on the McFarland scale and diluted in the ratio of either 1:20 (for rapidly growing mycobacteria) or 1:10 (for slow-growing mycobacteria) with broth.

The tested compounds were dissolved in DMSO (Sigma-Aldrich, Steinheim, Germany), then Middlebrook broth was added to obtain a concentration of 2000 µg/mL. The standards used for activity determination were INH, RIF, and CPX (Sigma-Aldrich, Steinheim, Germany). Final concentrations were reached by binary dilution and addition of mycobacterial suspension and were set as 500, 250, 125, 62.5, 31.25, 15.625, 7.81, and 3.91 µg/mL. INH was diluted in the range 500–3.91 µg/mL for screening against rapidly growing mycobacteria, 2000–15.625 µg/mL for *M. avium*, 50–0.39 µg/mL for *M. kansasii*, and 1–0.0078 µg/mL for *Mtb*. RIF final concentrations ranged from 50 to 0.39 µg/mL for rapidly growing mycobacteria, from 1 to 0.0078 µg/mL for *M. avium*, and from 0.1 to 0.00078 µg/mL for *Mtb* and *M. kansasii*. CPX was used for screening antimycobacterial activity with final concentrations of 1, 0.5, 0.25, 0.125, 0.0625, 0.0313, 0.0156, and 0.0078 µg/mL. The final concentration of DMSO did not exceed 2.5% (*v*/*v*) and did not affect the growth of any of the strains. Positive (broth, DMSO, bacteria) and negative (broth, DMSO) growth controls were included.

Plates containing slow-growing mycobacteria were sealed with polyester adhesive film and incubated in the dark at 37 °C without agitation. The addition of a 0.01% solution of resazurin sodium salt followed after 48 h of incubation for *M. smegmatis*, after 72 h for *M. aurum*, after 96 h for *M. avium* and *M. kansasii*, and after 120 h for *Mtb*. Microtitration panels were then incubated for a further 2.5 h to determine the activity against *M. smegmatis*, 4 h for *M. aurum*, 6 h for *M. avium* and *M. kansasii*, and 18 h for *Mtb*.

The antimycobacterial activity was expressed as minimal inhibition concentration (MIC) and the value was read on the basis of stain color change (blue color–active compound; pink color–inactive compound). MIC values for standards were in the range 15.625–31.25 µg/mL for INH, 12.5–25 µg/mL for RIF and 0.0625–0.125 µg/mL for CPX against *M. smegmatis*, 1.95–3.91 µg/mL for INH, 0.39–0.78 µg/mL for RIF, and 0.0078–0.0156 µg/mL for CPX against *M. aurum*, 500–1000 µg/mL for INH, 0.0625–0.125 µg/mL for RIF and 0.5 µg/mL for CPX against *M. avium*, 3.125–6.25 µg/mL for INH, 0.025–0.05 µg/mL for RIF and 0.25 µg/mL for CPX against *M. kansasii*, and 0.125–0.25 µg/mL for INH, 0.00156–0.0031 µg/mL for RIF and 0.125–0.25 for CPX against *Mtb*. All experiments were conducted in duplicate.

### 2.6. Cytotoxicity Assay

Human hepatocellular carcinoma HepG2 cells (ATCC HB-8065; passage 20–25), purchased from Health Protection Agency Culture Collections (ECACC, Salisbury, UK), were cultured in Minimum Essential Eagle Medium supplemented with 10% fetal bovine serum and 1% L-glutamine solution (Sigma-Aldrich, St. Louis, MO, USA) at 37 °C in a humidified atmosphere containing 5% CO_2_. For passage, the cells were treated with trypsin/EDTA (Sigma-Aldrich, St. Louis, MO, USA) at 37 °C and then harvested. For cytotoxicity evaluation, the cells treated with the test substances were used, while untreated HepG2 cells served as the control group; blank was included, as well. The cells were seeded in a 96-well plate at a density of 50,000 cells per well and incubated for 24 h. The following day, the cells were treated with each of the test compounds dissolved in DMSO (the highest DMSO concentration used was 0.5% *v*/*v*). The test substances were prepared at different concentrations in triplicates according to their solubility and incubated for 24 h in a humidified atmosphere containing 5% CO_2_ at 37 °C. After incubation, a solution of thiazolyl blue tetrazolium bromide (Sigma-Aldrich, St. Louis, MO, USA) in RPMI 1640 without phenol red (BioTech, Praha, Czech Republic) was added and incubated for 30 min in a humidified atmosphere containing 5% CO_2_ at 37 °C. Then, the formazan crystals formed were dissolved in DMSO and the absorbance of the samples was recorded at 570 nm (BioTek, Synergy Neo2 Multi-Mode Reader NEO2SMALPHAB). IC_50_ values were calculated by nonlinear regression from a semi-logarithmic plot of incubation concentration versus percentage of absorbance relative to untreated controls using GraphPad Prism software (version 9; GraphPad Software, Inc., La Jolla, CA, USA). The obtained results of the experiments are presented as the concentration that reduces the viability of cells from the maximal viability to 50% (IC_50_).

## 3. Results and Discussion

### 3.1. Isolation of Isoquinoline Alkaloids from Dicranostigma Franchetianum and Their Antimycobacterial Activity

Sixteen previously described IAs (**1**–**16**, Figure 1) has been isolated from the whole plant of *D. franchetianum* by standard chromatographic methods, as described in the Experimental section. By a combination of MS, ESI-HRMS, 1D and 2D NMR experiments, optical rotation, and by comparison of the obtained data with the literature, the compounds were identified as dihydrosanquinarine (**1**) [31], dihydrochelerythrine (**2**) [31], 6-ethoxydihydrochelerythrine (**3**) [32], stylopine (**4**) [33], *bis*-[6-(5,6-dihydrochelerythrinyl)]ether (**5**) [34], protopine (**6**) [35], allocryptopine (**7**) [33], cryptopine (**8**) [35], chelidonine (**9**) [36], isocorydine (**10**) [37], glaucine (**11**) [38], corydine (**12**) [37], isocorypalmine (**13**) [39], laudanosine (**14**) [40], scoulerine (**15**) [41] and berberine (**16**) [42]. The isolated IAs belong to the benzophenanthridine (**1**–**3**, **9**), protoberberine (**4**, **13**, **15**, **16**), bisbenzophenanthridine (**5**), protopine (**6**–**8**), aporphine (**10**–**12**), and simple benzylisoquinoline (**14**) structural types. 

Considering the anti-TB potency of alkaloids containing an isoquinoline heterocycle in their structure [43], the compounds isolated in sufficient amounts have been screened for their antimycobacterial potential against five *Mycobacterium* or *Mycolicibacterium* strains (Table 1). Most of the tested alkaloids showed either weak or no antimycobacterial potency (MIC ≥ 125 µg/mL, Table 1). Alkaloids **3** and **5**, which contain a benzophenanthridine skeleton in their structure, displayed moderate activity against all the tested mycobacterial strains (MICs 15.625–31.25 µg/mL, Table 1). However, alkaloid **3** was recognized as an isolation artifact only. According to the isolation conditions, chelerythrine was unintentionally introduced to ethanol in an acidic environment at the very beginning of the phytochemical work, where the iminium electrophile was readily trapped by a nucleophile. Thus, this nucleophilic addition furnished a hemiaminal ether **3**. Compound **5** is a dimer of two molecules of dihydrochelerythrine connected at position C-6 by an ether bridge and has been isolated previously from *Zanthoxylum paracanthum* and *Z. monophyllum* [44,45]. Dostál et al. have previously reported chelerythrine-like compounds in a thorough description of the dimerization. We assume that the action of the basic aqueous conditions of the phytochemical process yielded this dimer **5** [46]. Due to the induced dimerization, chelerythrine has never appeared in our final steps of isolation. Nevertheless, compound **5** exhibited strong activity against *Aspergillus fumigatus* (IC_50_ = 0.9 µM) and methicillin-resistant *Staphylococcus aureus* (IC_50_ = 1 µM) [45]. The improved antimycobacterial activity of **3** and **5**, compared to dihydrochelerythrine itself (**2**), can be an inspiration for the preparation of more potent antimycobacterial compounds derived from the benzophenanthridine skeleton.

In previous studies, berberine (**16**) exerted potent antimycobacterial activity against *M. intracellulare* with a MIC value of 1.56 µg/mL [47], and moderate activity against *Mycolicibacterium smegmatis* (MIC 25 µg/mL) [48]. On the contrary, the study of Gentry et al. 1998 disclosed a weak activity of **16** against *M. intracellulare* (MIC 200 µg/mL). For this reason, we revised the data for berberine (**16**) when challenged with other *Mycobacterium* strains; the compound demonstrated moderate activity against *M. avium* and *M. kansasii* with MIC values of 31.25 µg/mL against both strains (Table 1). Semisynthetic derivatives of **16** are reported as compounds with strong antimycobacterial potential, e.g., 2,3,9-triethoxy-3,9-dibenzyloxy-10-methoxy-13-*n*-octylprotoberberine chloride displaying activity against drug-susceptible *Mtb* strain H37Rv with a MIC value of 0.125 µg/mL. Moreover, this berberine derivative showed an intriguing effect against RIF- and INH-resistant *Mtb*, which implies a new mechanism of action [27]. Because berberine (**16**) was isolated in quantity allowing preparation of its semisynthetic derivatives, we continued our study with the design and synthesis of C-9 substituted berberine analogues to inspect their antimycobacterial activity.

### 3.2. Berberine Derivatives *(**16a**–**16k**)* and Their Antimycobacterial Activity

Berberine is currently one of the most studied NPs, with a wide range of biological activities, including antimicrobial, hepatoprotective, antihyperlipidemic, anticancer, antidiabetic, anti-inflammatory, and antiarrhythmic effects [49,50,51,52,53]. It is available globally, mainly as a root extract, providing antimicrobial and antidiabetic effects. Recent studies have shown that berberine can enhance the inhibitory efficiency of antibiotics against clinically multi-drug resistant isolates of *Staphylococcus aureus* [54]. Besides, berberine administration to mice with pulmonary tuberculosis resulted in decreased lung pathology with no additive or synergistic effects on bacterial burden [55]. Berberine has low permeability through biological membranes and poor bioavailability, limiting its application and further development. Structural modification of NPs is a powerful method to improve their pharmacological properties.

For this reason, structural modifications of berberine (**16**) that increase its lipophilicity, mostly at the C-9 position, have been pursued to address these drawbacks [28,56]. Based on our previous report on the antimycobacterial activity of semisynthetic derivatives of Amaryllidaceae alkaloids, which showed promising activity against three *Mycobacterium* strains, we selected berberine (**16**) for the preparation of a small library of its semisynthetic derivatives, which have been screened for antimycobacterial activity against five *Mycobacterium* strains (Table 1). Given the necessity of long-term administration of anti-TB drugs, we have also determined their potential hepatotoxicity in vitro using a hepatocellular HepG2 cell line.

Derivatives (**16a**–**16k**; Figure 2) of berberine (**16**) were synthesized using the same method as reported by Sobolová et al., 2020 [28].

In all cases, replacing the methoxy group at position C-9 by differently substituted benzyl appendages was associated with a significant increase in antimycobacterial activity against all studied *Mycobacterium* strains (MIC 0.39–7.81 µg/mL, Table 1). For all the tested compounds, *Mycolicibacterium smegmatis* was the most sensitive strain. Among the methyl-substituted benzylberberine derivatives (**16b**–**16d**), a *para* position of the methyl group on the aromatic ring yielded the most active compound. Bromine substitution within the benzyl moiety resulted in better activity against all strains, generating the most active compound within the study 9-*O*-(2-brombenzyl)berberine (**16e**) (MIC 0.390–3.125 µg/mL). For this halogen substituent, the *ortho* position at the aromatic ring was connected with the highest activity. 3,4-Dichloro-substituted analogue **16k** showed the same activity against *Mtb* as **16e**, and either the same or slightly lower potency against the other tested strains (Figure 2, Table 1).

### 3.3. Lipophilicity versus Activity

Lipophilicity is one of the most important physicochemical properties of a compound that is crucially related to cell transmembrane transport. The mycobacterial cell wall is rich in mycolic acids, a distinctive feature of the *mycobacterial* cell wall, which efficiently prevents the penetration of drugs. Drugs with higher lipophilicity may exert better activity against *Mtb* [57], and thus we calculated ClogP values using the ChemBioDraw Ultra program (ver 18.1; PerkinElmer, Waltham, MA, USA). The ClogP value is correlated directly to the molecular hydrophobicity, presuming the diffusion through the biological membranes. All synthesized derivatives showed a higher ClogP value (0.99–2.30) in comparison with parent berberine (−0.77).

### 3.4. Cytotoxicity of Berberine Derivatives

Compounds with MIC values lower than 20 µM for *Mtb* H37Ra were also evaluated for in vitro cytotoxicity on hepatocellular carcinoma cells (HepG2) using an MTT assay. The HepG2 cell line serves as an in vitro model for hepatotoxicity for early drug screening. Moreover, the hepatocellular model was chosen since anti-TB drugs are known to carry the risk of hepatotoxicity, mostly ascribed to their long-term administration [58].

Evaluation of cytotoxicity allows the calculation of selectivity indexes (SI) as the ratio of IC_50_, HepG2 to MIC of *Mtb* H37Ra (Table 1; Appendix A). In general, values of SI higher than 10 indicate more acceptable toxicity (analogous to the therapeutic index). The active compounds were screened at an initial concentration of 50 µM, and the IC_50_ values were subsequently determined. The effective cytotoxic concentrations of all tested compounds are listed in Table 1.

Among berberine derivatives, compound **16h** exhibited the highest cytotoxicity with an IC_50_ value of 9.44 ± 0.29 µM. The most active derivatives in terms of the highest antimycobacterial potency, i.e., **16e** and **16k**, showed cytotoxicity, with IC_50_ values of 11.61 ± 0.29 µM, and 12.66 ± 2.51 µM, respectively, reaching SI values of 3.92 for **16e**, and 4.55 for **16k**. These values indicate a potential risk of hepatotoxicity. Thus, further research to find a balance between antimycobacterial efficacy and hepatotoxicity will be necessary to optimize the berberine scaffold.

## 4. Conclusions

In conclusion, the phytochemical investigation of the alkaloidal extract of the whole plant of *Dicranostigma franchetianum* allowed the isolation of 16 known isoquinoline alkaloids of different structural types. Compounds isolated in sufficient amounts were evaluated for their in vitro antimycobacterial activity against *Mtb* H37Ra and four other mycobacterial strains of clinical importance or usually used in antimycobacterial screening (*Mycolicibacterium aurum*, *Mycobacterium avium*, *Mycobacterium kansasii*, and *Mycolicibacterium smegmatis*). 6-Ethoxydihydrochelerythrine and *bis*-[6-(5,6-dihydrochelerythrinyl)]ether, both containing a benzophenanthridine nucleus in their structure, demonstrated moderate activity against all tested strains. The strong antimycobacterial potential of berberine, reported in previous studies, was not confirmed. 

On the other hand, its semisynthetic derivatives demonstrated significant activity increment against all tested mycobacterial strains. Unfortunately, the most active berberine derivatives showed a relatively high cytotoxicity profile, as determined on HepG2 cell lines, indicating a potential hepatotoxicity risk. Therefore, further research on the berberine scaffold will focus on developing novel agents retaining high antimycobacterial profile while also reducing the cytotoxicity of the final entity. Further research warrants testing the top-ranked drug candidates in a combinatorial regimen by checkerboard studies to determine their synergistic properties with known antitubercular drugs. In vivo toxicity and in vivo efficacy will be determined using *Galleria mellonella* animal model. As promising antimycobacterial compounds, this pilot study presents prospective C-9 substituted berberine derivatives. Nevertheless, structure optimization needs further development and endeavor.

## Figures and Tables

**Figure 1 biomolecules-12-00844-f001:**
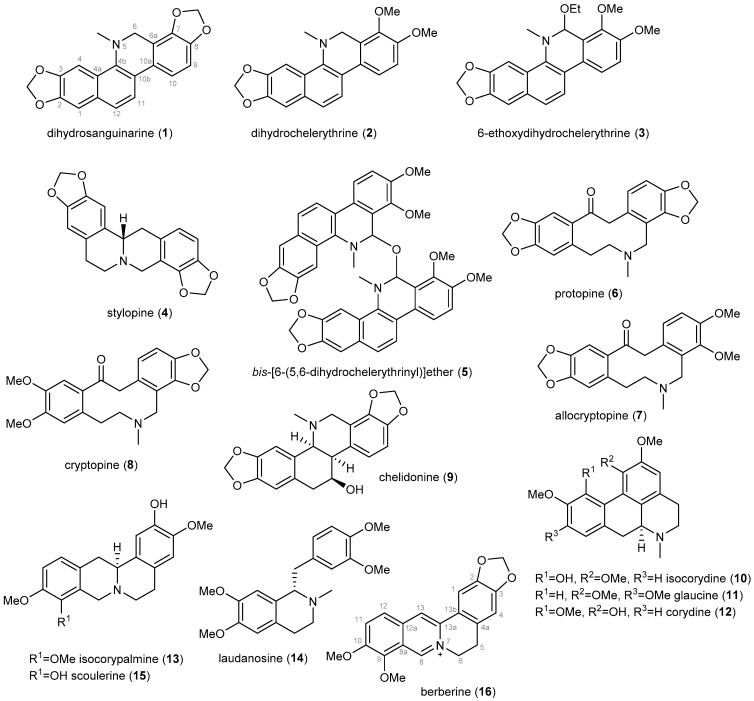
Isoquinoline alkaloids isolated from the whole plant of *Dicranostigma franchetianum*.

**Figure 2 biomolecules-12-00844-f002:**
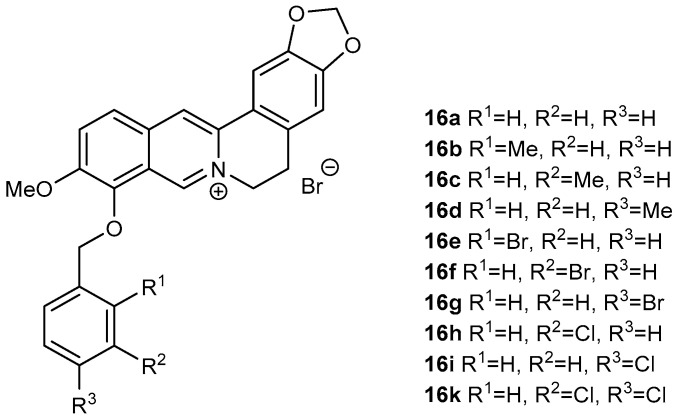
Berberine derivatives from our library investigated as antitubercular agents within this study [28].

**Table 1 biomolecules-12-00844-t001:** In vitro antimycobacterial activity against *Mtb* H37Ra, *Mycolicibacterium aurum*, *Mycobacterium avium*, *Mycobacterium kansasii*, and *Mycolicibacterium smegmatis* (MIC), cytotoxicity (IC_50_), selectivity index (SI), and calculated lipophilicity (ClogP) of isoquinoline alkaloids and berberine derivatives.

Alkaloid/Derivative	*Mtb* H37Ra (µg/mL)	*Mtb* H37Ra (µM) ^a^	*M. aurum* (µg/mL)	*M. avium* (µg/mL)	*M. kansasii* (µg/mL)	*M. smegmatis* (µg/mL)	HepG2 IC_50_ (µM)	SI ^b^	ClogP ^c^
dihydrosanguinarine (**1**)	≥500	≥1501	≥500	≥500	125	≥500	n.s.	n.c.	5.23
dihydrochelerythrine (**2**)	250	716	250	62.5	250	≥500	n.s.	n.c.	4.92
6-ethoxydihydrochelerythrine (**3**)	31.25	79	15.625	31.25	31.25	31.25	n.s.	n.c.	5.47
stylopine (**4**)	≥125	≥387	≥125	≥125	≥125	≥125	n.s.	n.c.	3.81
*bis*-[6-(5,6-dihydrochelerythrinyl)]ether (**5**)	31.25	44	31.25	31.25	31.25	31.25	n.s.	n.c.	8.89
protopine (**6**)	≥500	≥1416	≥500	250	120	≥500	n.s.	n.c.	3.57
allocryptopine (**7**)	≥250	≥678	≥250	≥250	250	≥250	n.s.	n.c.	3.48
cryptopine (**8**)	≥125	≥339	≥125	≥125	≥125	≥125	n.s.	n.c.	3.48
isocorydine (**10**)	≥500	≥1432	≥500	≥500	125	≥500	n.s.	n.c.	2.60
glaucine (**11**)	125	≥352	125	125	62.5	≥250	n.s.	n.c.	3.07
corydine (**12**)	≥500	≥1466	250	62.5	250	≥500	n.s.	n.c.	2.82
isocoryplamine (**13**)	≥250	≥733	≥250	≥250	≥250	≥250	n.s.	n.c.	2.72
scoulerine (**15**)	250	764	125	250	15.625	125	n.s.	n.c.	2.25
berberine (**16**)	125	336	62.5	31.25	31.25	62.5	n.s.	n.c.	−0.77
9-*O*-benzylberberine chloride (**16a**)	6.25	13.9	3.125	3.125	6.25	0.78	24.47 ± 4.25	1.76	0.99
9-*O*-(2-methylbenzyl)berberine chloride (**16b**)	7.81	16.9	7.81	3.91	3.91	1.95	39.27 ± 3.62	2.32	1.45
9-*O*-(3-methylbenzyl)berberine chloride (**16c**)	6.25	13.5	3.125	6.25	6.25	1.56	47.13 ± 7.38	3.49	1.49
9-*O*-(4-methylbenzyl)berberine chloride (**16d**)	3.91	8.46	6.25	1.98	3.91	1.56	15.86 ± 3.48	2.54	1.49
9-*O*-(2-brombenzyl)berberine chloride (**16e**)	**1.56**	**2.96**	**0.78**	**1.56**	**3.125**	**0.39**	**11.75 ± 0.29**	**3.92**	**1.86**
9-*O*-(3-brombenzyl)berberine chloride (**16f**)	3.125	5.93	1.56	3.125	3.125	0.78	13.90 ± 3.25	2.34	1.86
9-*O*-(4-brombenzyl)berberine chloride (**16g**)	3.91	7.42	3.91	1.98	0.98	0.98	21.44 ± 4.80	2.89	1.86
9-*O*-(3-chlorbenzyl)berberine chloride (**16h**)	3.125	5.93	1.56	1.56	3.125	0.78	9.44 ± 2.13	1.59	1.71
9-*O*-(4-chlorbenzyl)berberine chloride (**16i**)	3.125	5.93	3.125	1.56	6.25	1.56	16.17 ± 1.67	2.73	1.71
9-*O*-(3,4-dichlorbenzyl)berberine chloride (**16k**)	**1.56**	**2.78**	**1.56**	**1.56**	**6.25**	**1.56**	**12.66 ± 2.51**	**4.55**	**2.30**
isoniazid ^d^	0.25	1.82	3.91	500	6.25	31.25	n.s.	n.c.	−0.67
rifampicin ^d^	0.00625	0.0075	0.39	0.125	0.025	12.5	n.s.	n.c.	3.71
ciprofloxacin ^d^	0.25	0.75	0.015625	0.5	0.25	0.0625	n.s.	n.c.	−0.62
doxorubicine	ns	ns	ns	ns	ns	ns	30.38 ± 1.74	n.c.	n.c.

^a^ Calculated from MIC (µg/mL) and MW, ^b^ SI–Selectivity index, values calculated from MIC against *Mtb* H37Ra (in µM) and IC_50_ HepG2/MIC (in µM), ^c^ LogP and CLogP calculated in ChemDraw v18.1.; ^d^ standard; n.s. stands for not studied; n.c. stands for not calculated.

## Data Availability

The data presented in this study are available on request from the corresponding author.

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
