# Peer review of "Alkaloids of Dicranostigma franchetianum (Papaveraceae) and Berberine Derivatives as a New Class of Antimycobacterial Agents"

_biomolecules, 2022, doi:10.3390/biom12060844_

Round 1

Reviewer 1 Report

In this manuscript, Viriyanata Wijaya et al reported three parts of work, including 1) Isolation of isoquinoline alkaloids and semi-synthesis of 10 berberine derivatives (their previous work); 2) The anti- mycobacterial activity study of the compounds; and 3) The hepatotoxicity study of some berberine derivatives with anti- mycobacterial activity.

There are some flaws in the work which should be modified before it is published:

1, In anti-mycobacterial study , the authors used 5 mycobacterial strains and got different MIC for a certain compounds. Why to use such different bacterial strains and what the results of the different MIC for different strains suggest should be discussed in the manuscript.

2, In the hepatotoxicity study , 1) a positive control should be used. 2) And why the authors used 16 -hour treatment for this study? Usually, such study uses 48- hour or 72- hour treatment. 3) The IC50 data in the manuscript are not consistent with the data in the file of non-published, as as IC50 for 16e. And no data of IC50 for 16a.

3, Among IAs isolated from the plant, alkaloids 3 and 5 were found to have moderate antimycobacterial activity. While the authors discussed alkaloid 3 was recognized as an isolation artifact. While compound 5 is a dimer of two molecules of dihydrocheletythrine, whether it is from the plant or from compound 3 required more study.

4, The manuscript lacks the rigorous logic to design the experiment, describe and discuss the results. For example, in the background, the authors stressed the importance of new drug with new backbones against drug resistant TB, while in the anti-mycobaterial study, they did not focus on the drug resistance and new backbones. The structure-activity relationship has not been studied enough in the work. Moreover, the authors studied the lipophilicity of berberine derivatives, while lacked extensive discussion on the relation of lipophilicity vs. activity, since the positive control drugs with low ClogP while high anti-mycobacterial activity.

Other minor errors:

1, Some results were described in the method section(P 5, Line 236-244) and some methods were descrbed in the result section(P10, 3.3 Lipophilicity vs.activity).

2, The concentrate unit should be consistent. It is better to use Molar concentration. µM/mL in “aCalculated from MIC (µM/mL)” was wrong.

3, P6, L255, “no cell control was included” what does it mean?

4, P3, L137, “Botanical identification was performed by Prof. L. Opletal” should be described through which way to identify it.

5, In “2.5 Antimycobacterial screening” , the authors used color-based MIC study and since no reference paper was cited , photos with blue (active ) and pink (inactive) had better to attached as supplementary data.

6 P10, L46-52 should not be in the conclusion section. It should be in the discussion section.

Author Response

Biomolecules

Dear Editor,

We have carefully revised manuscript titled " Alkaloids of Dicranostigma franchetianum (Papaveraceae) and berberine derivatives as a new class of antimycobacterial agents" Manuscript ID: biomolecules-1750593, according to the reviewers' suggestions. All changes are highlighted in red.

Comments from the reviewers:

Reviewer #1:

In this manuscript, Viriyanata Wijaya et al reported three parts of work, including 1) Isolation of isoquinoline alkaloids and semi-synthesis of 10 berberine derivatives (their previous work); 2) The anti- mycobacterial activity study of the compounds; and 3) The hepatotoxicity study of some berberine derivatives with anti- mycobacterial activity.

There are some flaws in the work which should be modified before it is published:

1, In anti-mycobacterial study , the authors used 5 mycobacterial strains and got different MIC for a certain compounds. Why to use such different bacterial strains and what the results of the different MIC for different strains suggest should be discussed in the manuscript.

The list of mycobacterial species is chosen according to recommendations and procedures typically used for antimycobacterial compounds screening and also with regard to epidemiological situation. M. smegmatis and M. aurum as fast-growers are used in research of novel compounds as surrogate model with low risk of infection and there are also possible correlations between activity against these strains and M. tuberculosis. Other two strains (M. avium and M. kansasii) are nowadays considered to be of clinical importance because the incidence of so called mycobacterioses is rising, especially in immunocompromised patients.

The differences in MIC values obtained can be explained by subtle differences in enzymatic repertoire of these mycobacterial strains. But results in this paper showed similar results for all strains across all compounds. The differences are in range of one dilution showing not just antitubercular but antimycobacterial effect against mycobacteria in general (positive effect).

2, In the hepatotoxicity study , 1) a positive control should be used. 2) And why the authors used 16 -hour treatment for this study? Usually, such study uses 48- hour or 72- hour treatment. 3) The IC50 data in the manuscript are not consistent with the data in the file of non-published, as as IC50 for 16e. And no data of IC50 for 16a.

3, Among IAs isolated from the plant, alkaloids 3 and 5 were found to have moderate antimycobacterial activity. While the authors discussed alkaloid 3 was recognized as an isolation artifact. While compound 5 is a dimer of two molecules of dihydrocheletythrine, whether it is from the plant or from compound 3 required more study.

Ethoxy-derivatives from natural products are not common types of secondary metabolites. They can be created during isolation process when EtOH is used during separation. (Berkov, S.; Osorio, E.; Viladomat, F.; Bastida, J., Chapter Two - Chemodiversity, chemotaxonomy and chemoecology of Amaryllidaceae alkaloids. In The Alkaloids: Chemistry and Biology, Knölker, H.-J., Ed. Academic Press: 2020; Vol. 83, pp 113-185).

4, The manuscript lacks the rigorous logic to design the experiment, describe and discuss the results. For example, in the background, the authors stressed the importance of new drug with new backbones against drug resistant TB, while in the anti-mycobaterial study, they did not focus on the drug resistance and new backbones. The structure-activity relationship has not been studied enough in the work. Moreover, the authors studied the lipophilicity of berberine derivatives, while lacked extensive discussion on the relation of lipophilicity vs. activity, since the positive control drugs with low ClogP while high anti-mycobacterial activity.

Experiments with resistant strains is more demanding. It is considered only for compounds substances with excellent efficacy (MIC below 1 μM). Due to the fact that our compounds showed relatively low SI, further series of berberine derivatives will be prepared in follow-up experiments. If any of them show SI higher than 10, it will also be tested for resistant strains.

Other minor errors:

1, Some results were described in the method section(P 5, Line 236-244) and some methods were descrbed in the result section(P10, 3.3 Lipophilicity vs.activity).

In section 3.3 Lipophilicity vs.activity is mentioned only short introduction into lipophilicity and why we calculated it.

2, The concentrate unit should be consistent. It is better to use Molar concentration. µM/mL in “aCalculated from MIC (µM/mL)” was wrong.

Corrected

3, P6, L255, “no cell control was included” what does it mean?

We wanted to mention that we include a blank in each measurement for values correction. We modified the text in the manuscript accordingly.

4, P3, L137, “Botanical identification was performed by Prof. L. Opletal” should be described through which way to identify it.

A good specimen, for accurate identification, include several leaves attached to a section of stem, identification of flowers or fruits, and if possible roots. Expression „Botanical identification“ is widely used in phytochemical studies, we think it is not necessary to explain in detail in the present manuscript.

5, In “2.5 Antimycobacterial screening” , the authors used color-based MIC study and since no reference paper was cited , photos with blue (active ) and pink (inactive) had better to attached as supplementary data.

Citations of reference paper was added in manuscript.

6, P10, L46-52 should not be in the conclusion section. It should be in the discussion section.

We have left this part in its original place, as it outlines the continuation of the entire research. In our opinion, it does not fit into any parts of the discussion.

Reviewer 2

The paper by Wijaya et al. describes the isolation of alkaloids from the Chinese plant Dicranostigma franchetianum and illustrates their biological analysis on a panel of different mycobacterial species. The authors also report the synthesis of a small library of berberine derivatives, inspired by the modest activity of this natural compound on some mycobacteria. Finally, they evaluate the cytotoxicity of the synthesised compounds on liver HepG2 cells, revealing a potential toxicity for the most active candidates.

The paper appears to be simple and straightforward in its conceptualisation. The results are not particularly relevant from a medicinal chemistry point of view, mainly due to the toxicity of the compounds and to the lack of a defined molecular target. Nonetheless, the work appears to be scientifically sound and quite complete. It is the opinion of this reviewer that rigorous scientific results are worthy of being published, independently of the success of the proposed approach. Hence, the paper could be published, but some issues need to be addressed first.

A moderate/extensive revision of the English is necessary. Despite concepts are generally clear, the phrasing is sometimes awkward (e.g., page 1 lines 28-29; page 2 lines 96-97; page 6, line 287; page 10, lines 3-4; pages 10-11, lines 49-52).

We checked and rephrased these sentences.

The Introduction section on TB is quite exhaustive, but some references should be added regarding the role of NPs as antitubercular agents. An interesting review on this topic was recently published by Cazzaniga et al. on the European Journal of Medicinal Chemistry (https://doi.org/10.1016/j.ejmech.2021.113732).

Reference has been incorporated into manuscript.

The significance of the section dedicated to berberine derivatives should be better illustrated. The authors reported a considerable amount of berberine analogues in a previous work, published on the European Journal of Medicinal Chemistry in 2020. Most notably, they had already detected a potential toxicity related to the use of these compounds. Hence, this reviewer wonders about the significance of adding a handful of already published derivatives to this work. A better contextualisation is highly recommended. Moreover, a brief, general description of the synthetic procedure should be added. Scheme 1 should also be revised and clarified. For example, where does the chloride counterion come from in the product of the first reaction? The starting reagent is probably a chloride salt (?). Moreover, how can the phenol OH be deprotonated (where is the counterion?)? The deprotonation should be the result of the presence of the base in the following substitution reaction.

We decided to change Scheme 1 to Figure 2, simplifying the description and omitting the chemical synthesis as it has already been reported in Babkova, K. et al. Eur. J. Med. Chem. 2020. The significance of adding other berberine derivatives into the study stems from the fact these types of compounds revealed antitubercular activity, as reported before (e.g. doi: 10.1007/s10600-014-0942-8; doi: 10.1016/j.apsb.2012.10.008); thus broadening the knowledge of berberine-related compounds.

A recent article was published by Ozturk et al. on Frontiers in Immunology (https://doi.org/10.3389/fimmu.2021.656419) regarding the use of berberine as an adjuvant in TB treatment. The discussion on berberine and its derivatives could be implemented also considering their findings, especially in the framework of further potential uses of this scaffold. The reviewer encourages to widen the literature search to draw some conclusion on the use of berberine derivatives in anti-TB drug discovery.

Thank you for noticing. We have mentioned the study discussing the potential of berberine administration to pulmonary form of tuberculosis.

Some minor issues should also be resolved:

Abstract, lines 38-41: please clarify, how can “lower” IC50 values on HepG2 cells with respect to the MICs corroborate the safety of the compounds?

This issue is already addressed in chapter 3.4, lines 19-23. If the IC50 for the cytotoxicity is significantly lower than the MIC, the compound will reach an effective concentration without triggering the toxic side effect. Our evaluation using only HepG2 cell lines is, of course, very tentative, and to more accurately assess the toxicity/effectivity ratio, further experiments are needed. 

The clarification was incorporated into the abstract.

When reporting and confronting numerical data, the same number of decimal figures should be used for all values (e.g., page 1, line 34). This issue is frequently repeated throughout the whole text, and, most notably, in Table 1. Moreover, in Table 1, the standard deviations are indicated only for IC50 values against HepG2 cells; why is that?

We used standard dilution-micrometod using two-fold dilution. The results for.M.tuberculosis are also experresed in MIC for calculation of SI. We expressed our results in the usual way so that the results can be compared with other data ( doi: 10.1016/j.jpha.2015.11.005; DOI: 10.1016/j.ejmech.2012.03.012;  DOI: 10.1016/j.ejmech.2019.111578

It is desirable to use the same unit of measure when reporting MIC values (μg/mL or μM). Discrepancies should be commented upon. Also, it would be preferrable to specify if the reported values are MIC50 or MIC99.

Standard methodologies (according to international recommendations – e.g. EUCAST) prefer values in µg/mL, so we also measured MIC in this unit. But for calculations of SI, we have calculated also MIC value for Mtb in µM using molecular weight.

Value of MIC is usually taken as 95-99% of inhibition. Different percentages would be highlighted as IC with lower index number corresponding to percentage of inhibition.

The use of units of measure should be revised throughout the text. Numbers and “°C” should be separated by a space when temperatures are reported. “L” should be used instead of “litres” (page 3, line 145). “h” should be used instead of “hours” (page 5, lines 231-235).

Corrected

When acronyms are defined, it is unnecessary to repeat the definition multiple times (page 5, line 217 ® page 7, lines 317-318).

Corrected

Page 7, line 316: “H37Rv” should be “H37Rv”.

Corrected

Page 7, Figure 1: because the names of the molecules are all indicated under the structures, it may be better to adjust the position of the name of compound 14 to avoid confusion.

Corrected

Page 8, line 356: “3,4-Dichlorine-substituted” should be replaced by “3,4-Dichloro-substituted”.

Corrected

Page 10, chapter 3.3: at lines 9-10, the reviewer suggests striking “i.e. through the mycobacterial cell wall”. Moreover, the reviewer also recommends striking or clarifying the reference to the greater lipophilicity of the berberine derivatives with respect to known drugs, because it seems to imply that they should be more active than the established therapeutic agents, while, in fact, many other factors other than lipophilicity contribute to determine antimycobacterial activities.

Corrected as recommended (striked).

References should be revised to include (where possible) missing DOIs and issues/pages.

According to the journal reference style, issue numbers are neither required nor used in recent Biomolecules' articles. Where available, the DOIs and pages were supplemented.

Thank you very much for your valuable feedback and time spent reviewing the manuscript.

In light of these changes, we are positive that our revised manuscript meets the criteria to be published in Biomolecules and would be of interest for all readers from the scientific community, with particular emphasis on drug development against mycobacterial infections.

Lucie Cahlikova

Reviewer 2 Report

The paper by Wijaya et al. describes the isolation of alkaloids from the Chinese plant Dicranostigma franchetianum and illustrates their biological analysis on a panel of different mycobacterial species. The authors also report the synthesis of a small library of berberine derivatives, inspired by the modest activity of this natural compound on some mycobacteria. Finally, they evaluate the cytotoxicity of the synthesised compounds on liver HepG2 cells, revealing a potential toxicity for the most active candidates.

The paper appears to be simple and straightforward in its conceptualisation. The results are not particularly relevant from a medicinal chemistry point of view, mainly due to the toxicity of the compounds and to the lack of a defined molecular target. Nonetheless, the work appears to be scientifically sound and quite complete. It is the opinion of this reviewer that rigorous scientific results are worthy of being published, independently of the success of the proposed approach. Hence, the paper could be published, but some issues need to be addressed first.

·       A moderate/extensive revision of the English is necessary. Despite concepts are generally clear, the phrasing is sometimes awkward (e.g., page 1 lines 28-29; page 2 lines 96-97; page 6, line 287; page 10, lines 3-4; pages 10-11, lines 49-52).

·                 ●  The Introduction section on TB is quite exhaustive, but some references should be added regarding the role of NPs as antitubercular agents. An interesting review on this topic was recently published by Cazzaniga et al. on the European Journal of Medicinal Chemistry (https://doi.org/10.1016/j.ejmech.2021.113732).

·              ●  The significance of the section dedicated to berberine derivatives should be better illustrated. The authors reported a considerable amount of berberine analogues in a previous work, published on the European Journal of Medicinal Chemistry in 2020. Most notably, they had already detected a potential toxicity related to the use of these compounds. Hence, this reviewer wonders about the significance of adding a handful of already published derivatives to this work. A better contextualisation is highly recommended. Moreover, a brief, general description of the synthetic procedure should be added. Scheme 1 should also be revised and clarified. For example, where does the chloride counterion come from in the product of the first reaction? The starting reagent is probably a chloride salt (?). Moreover, how can the phenol OH be deprotonated (where is the counterion?)? The deprotonation should be the result of the presence of the base in the following substitution reaction.

·         ●  A recent article was published by Ozturk et al. on Frontiers in Immunology (https://doi.org/10.3389/fimmu.2021.656419) regarding the use of berberine as an adjuvant in TB treatment. The discussion on berberine and its derivatives could be implemented also considering their findings, especially in the framework of further potential uses of this scaffold. The reviewer encourages to widen the literature search to draw some conclusion on the use of berberine derivatives in anti-TB drug discovery.

Some minor issues should also be resolved:

·           ● Abstract, lines 38-41: please clarify, how can “lower” IC50 values on HepG2 cells with respect to the MICs corroborate the safety of the compounds?

·                   ●  When reporting and confronting numerical data, the same number of decimal figures should be used for all values (e.g., page 1, line 34). This issue is frequently repeated throughout the whole text, and, most notably, in Table 1. Moreover, in Table 1, the standard deviations are indicated only for IC50 values against HepG2 cells; why is that?

·             ● It is desirable to use the same unit of measure when reporting MIC values (μg/mL or μM). Discrepancies should be commented upon. Also, it would be preferrable to specify if the reported values are MIC50 or MIC99.

·             ●  The use of units of measure should be revised throughout the text. Numbers and “°C” should be separated by a space when temperatures are reported. “L” should be used instead of “litres” (page 3, line 145). “h” should be used instead of “hours” (page 5, lines 231-235).

·               ●  When acronyms are defined, it is unnecessary to repeat the definition multiple times (page 5, line 217 ® page 7, lines 317-318).

·                   ●  Page 7, line 316: “H37Rv” should be “H37Rv”.

·                ●  Page 7, Figure 1: because the names of the molecules are all indicated under the structures, it may be better to adjust the position of the name of compound 14 to avoid confusion.

·            ●   Page 8, line 356: “3,4-Dichlorine-substituted” should be replaced by “3,4-Dichloro-substituted”.

·            ●  Page 10, chapter 3.3: at lines 9-10, the reviewer suggests striking “i.e. through the mycobacterial cell wall”. Moreover, the reviewer also recommends striking or clarifying the reference to the greater lipophilicity of the berberine derivatives with respect to known drugs, because it seems to imply that they should be more active than the established therapeutic agents, while, in fact, many other factors other than lipophilicity contribute to determine antimycobacterial activities.

·          ●  References should be revised to include (where possible) missing DOIs and issues/pages.

Author Response

(The authors gave the same response as above.)

Round 2

Reviewer 1 Report

The authors have responsed to my comments. It is better to add the decription to the manuscipt, to help  the readers understand better, such as the the bacterial strain choice, the time point choice. 

Author Response

Biomolecules

Dear Editor,

We have carefully revised manuscript titled " Alkaloids of Dicranostigma franchetianum (Papaveraceae) and berberine derivatives as a new class of antimycobacterial agents" Manuscript ID: biomolecules-1750593, according to the reviewers' suggestions. All changes are highlighted in red.

Comments from the reviewers:

Reviewer #1:

The authors have responsed to my comments. It is better to add the description to the manuscript, to help the readers understand better, such as the the bacterial strain choice, the time point choice.

We thank the reviewers for their comments and have revised.

Reviewer #2:

The revised paper by Wijaya et al. appears to be improved with respect to the previous version in terms of clarity and structure. Most of the raised issues have been dealt with properly. However, this reviewer continues to recommend further language polishing (e.g., page 1 line 28: using the plural “drugs” would be better; page 2, line 97: the use of “whereas” makes the sentence unclear; page 8, line 327: the parenthesis is formally ambiguous, meaning that the concept is clear, but the form can be improved; etc.). Hence, the paper is now ready to be published, after minor revisions. As for the specific issues raised in the review process, the reviewer would like to clarify some points and add some comments.

We incorporated recommendations into text.

In case of page 8, line 327, we removed informations in the parenthesis, as not to introduce ambiguity into the text.

We decided to change Scheme 1 to Figure 2, simplifying the description and omitting the chemical synthesis as it has already been reported in Babkova, K. et al. Eur. J. Med. Chem. 2020. The significance of adding other berberine derivatives into the study stems from the fact these types of compounds revealed antitubercular activity, as reported before (e.g. doi: 10.1007/s10600-014-0942-8; doi: 10.1016/j.apsb.2012.10.008); thus broadening the knowledge of berberine-related compounds.

The choice of removing the scheme is reasonable. The introduction to the importance of berberine derivatives has been improved by the added reference (Ozturk et al.). The authors rightfully point out in their answer that berberine derivatives have a documented antitubercular activity, but they do not explicitly cite the works in the paper. Why is that (have I missed them?)? Discussing the importance of berberine derivatives in anti-TB drug discovery by adding those references would greatly improve (and better contextualise) the introduction to the following results.

We added further references reporting antitubercular activity of berberine-derivatives.

This issue is already addressed in chapter 3.4, lines 19-23. If the IC50 for the cytotoxicity is significantly lower than the MIC, the compound will reach an effective concentration without triggering the toxic side effect. Our evaluation using only HepG2 cell lines is, of course, very tentative, and to more accurately assess the toxicity/effectivity ratio, further experiments are needed.

The correction to the abstract in the current form is acceptable. Originally, the results were duly discussed in chapter 3.4, but the sentence in the abstract was conceptually ambiguous. All HepG2 IC50 reported in Table 1 are higher than Mtb H37Ra MIC (e.g.16b: MIC = 16.9 μM; IC50 = 39.27 μM), which translates in a favourable SI. For a reference, see how Sikora and co-workers discuss the cytotoxicity of their peptides (10.1007/s00726-017-2536-9).

We used standard dilution-micrometod using two-fold dilution. The results for.M.tuberculosis are also experresed in MIC for calculation of SI. We expressed our results in the usual way so that the results can be compared with other data ( doi: 10.1016/j.jpha.2015.11.005; DOI: 10.1016/j.ejmech.2012.03.012; DOI: 10.1016/j.ejmech.2019.111578

Comparing results using unlike decimals remains debatable from a mathematical point of view. However, the reviewer realises that, in some cases, it is an issue intrinsic to the nature of the tests, so no further comments on the matter will be made.

Standard methodologies (according to international recommendations – e.g. EUCAST) prefer values in μg/mL, so we also measured MIC in this unit. But for calculations of SI, we have calculated also MIC value for Mtb in μM using molecular weight. Value of MIC is usually taken as 95-99% of inhibition. Different percentages would be highlighted as IC with lower index number corresponding to percentage of inhibition.

The reviewer’s intent was not to criticise the choice of calculating MIC values as either μg/mL or μM. However, when discussing the results, it would be better to use the same unit of measure (see page 1 lines 34, 37, 38). As for the specification of the MIC99, the reviewer agrees with the authors; the comment was purely meant as a suggestion, it is far from being a necessary modification.

According to the journal reference style, issue numbers are neither required nor used in recent Biomolecules' articles. Where available, the DOIs and pages were supplemented.

The authors are correct. The reference to the “issue number” was actually an oversight on the reviewer’s part. The indication was referred to the missing “volume” of reference 42 (Omosa et al.).

Thank you very much for your valuable feedback and time spent re-checking the manuscript.

In light of these changes, we are positive that our revised manuscript meets the criteria to be published in Biomolecules and would be of interest for all readers from the scientific community, with particular emphasis on drug development against mycobacterial infections.

Lucie Cahlikova

Reviewer 2 Report

The revised paper by Wijaya et al. appears to be improved with respect to the previous version in terms of clarity and structure. Most of the raised issues have been dealt with properly. However, this reviewer continues to recommend further language polishing (e.g., page 1 line 28: using the plural “drugs” would be better; page 2, line 97: the use of “whereas” makes the sentence unclear; page 8, line 327: the parenthesis is formally ambiguous, meaning that the concept is clear, but the form can be improved; etc.). Hence, the paper is now ready to be published, after minor revisions. As for the specific issues raised in the review process, the reviewer would like to clarify some points and add some comments.

We decided to change Scheme 1 to Figure 2, simplifying the description and omitting the chemical synthesis as it has already been reported in Babkova, K. et al. Eur. J. Med. Chem. 2020. The significance of adding other berberine derivatives into the study stems from the fact these types of compounds revealed antitubercular activity, as reported before (e.g. doi: 10.1007/s10600-014-0942-8; doi: 10.1016/j.apsb.2012.10.008); thus broadening the knowledge of berberine-related compounds.

The choice of removing the scheme is reasonable. The introduction to the importance of berberine derivatives has been improved by the added reference (Ozturk et al.). The authors rightfully point out in their answer that berberine derivatives have a documented antitubercular activity, but they do not explicitly cite the works in the paper. Why is that (have I missed them?)? Discussing the importance of berberine derivatives in anti-TB drug discovery by adding those references would greatly improve (and better contextualise) the introduction to the following results.

This issue is already addressed in chapter 3.4, lines 19-23. If the IC50 for the cytotoxicity is significantly lower than the MIC, the compound will reach an effective concentration without triggering the toxic side effect. Our evaluation using only HepG2 cell lines is, of course, very tentative, and to more accurately assess the toxicity/effectivity ratio, further experiments are needed.

The correction to the abstract in the current form is acceptable. Originally, the results were duly discussed in chapter 3.4, but the sentence in the abstract was conceptually ambiguous. All HepG2 IC50 reported in Table 1 are higher than Mtb H37Ra MIC (e.g., 16b: MIC = 16.9 μM; IC50 = 39.27 μM), which translates in a favourable SI. For a reference, see how Sikora and co-workers discuss the cytotoxicity of their peptides (10.1007/s00726-017-2536-9).

We used standard dilution-micrometod using two-fold dilution. The results for.M.tuberculosis are also experresed in MIC for calculation of SI. We expressed our results in the usual way so that the results can be compared with other data ( doi: 10.1016/j.jpha.2015.11.005; DOI: 10.1016/j.ejmech.2012.03.012; DOI: 10.1016/j.ejmech.2019.111578

Comparing results using unlike decimals remains debatable from a mathematical point of view. However, the reviewer realises that, in some cases, it is an issue intrinsic to the nature of the tests, so no further comments on the matter will be made.

Standard methodologies (according to international recommendations – e.g. EUCAST) prefer values in μg/mL, so we also measured MIC in this unit. But for calculations of SI, we have calculated also MIC value for Mtb in μM using molecular weight. Value of MIC is usually taken as 95-99% of inhibition. Different percentages would be highlighted as IC with lower index number corresponding to percentage of inhibition.

The reviewer’s intent was not to criticise the choice of calculating MIC values as either μg/mL or μM. However, when discussing the results, it would be better to use the same unit of measure (see page 1 lines 34, 37, 38). As for the specification of the MIC99, the reviewer agrees with the authors; the comment was purely meant as a suggestion, it is far from being a necessary modification.

According to the journal reference style, issue numbers are neither required nor used in recent Biomolecules' articles. Where available, the DOIs and pages were supplemented.

The authors are correct. The reference to the “issue number” was actually an oversight on the reviewer’s part. The indication was referred to the missing “volume” of reference 42 (Omosa et al.).

Author Response

Biomolecules

Dear Editor,

We have carefully revised manuscript titled " Alkaloids of Dicranostigma franchetianum (Papaveraceae) and berberine derivatives as a new class of antimycobacterial agents" Manuscript ID: biomolecules-1750593, according to the reviewers' suggestions. All changes are highlighted in red.

Comments from the reviewers:

Reviewer #2:

The revised paper by Wijaya et al. appears to be improved with respect to the previous version in terms of clarity and structure. Most of the raised issues have been dealt with properly. However, this reviewer continues to recommend further language polishing (e.g., page 1 line 28: using the plural “drugs” would be better; page 2, line 97: the use of “whereas” makes the sentence unclear; page 8, line 327: the parenthesis is formally ambiguous, meaning that the concept is clear, but the form can be improved; etc.). Hence, the paper is now ready to be published, after minor revisions. As for the specific issues raised in the review process, the reviewer would like to clarify some points and add some comments.

We incorporated recommendations into text.

In case of page 8, line 327, we removed informations in the parenthesis, as not to introduce ambiguity into the text.

We decided to change Scheme 1 to Figure 2, simplifying the description and omitting the chemical synthesis as it has already been reported in Babkova, K. et al. Eur. J. Med. Chem. 2020. The significance of adding other berberine derivatives into the study stems from the fact these types of compounds revealed antitubercular activity, as reported before (e.g. doi: 10.1007/s10600-014-0942-8; doi: 10.1016/j.apsb.2012.10.008); thus broadening the knowledge of berberine-related compounds.

The choice of removing the scheme is reasonable. The introduction to the importance of berberine derivatives has been improved by the added reference (Ozturk et al.). The authors rightfully point out in their answer that berberine derivatives have a documented antitubercular activity, but they do not explicitly cite the works in the paper. Why is that (have I missed them?)? Discussing the importance of berberine derivatives in anti-TB drug discovery by adding those references would greatly improve (and better contextualise) the introduction to the following results.

We added further references reporting antitubercular activity of berberine-derivatives.

This issue is already addressed in chapter 3.4, lines 19-23. If the IC50 for the cytotoxicity is significantly lower than the MIC, the compound will reach an effective concentration without triggering the toxic side effect. Our evaluation using only HepG2 cell lines is, of course, very tentative, and to more accurately assess the toxicity/effectivity ratio, further experiments are needed.

The correction to the abstract in the current form is acceptable. Originally, the results were duly discussed in chapter 3.4, but the sentence in the abstract was conceptually ambiguous. All HepG2 IC50 reported in Table 1 are higher than Mtb H37Ra MIC (e.g.16b: MIC = 16.9 μM; IC50 = 39.27 μM), which translates in a favourable SI. For a reference, see how Sikora and co-workers discuss the cytotoxicity of their peptides (10.1007/s00726-017-2536-9).

We used standard dilution-micrometod using two-fold dilution. The results for.M.tuberculosis are also experresed in MIC for calculation of SI. We expressed our results in the usual way so that the results can be compared with other data ( doi: 10.1016/j.jpha.2015.11.005; DOI: 10.1016/j.ejmech.2012.03.012; DOI: 10.1016/j.ejmech.2019.111578

Comparing results using unlike decimals remains debatable from a mathematical point of view. However, the reviewer realises that, in some cases, it is an issue intrinsic to the nature of the tests, so no further comments on the matter will be made.

Standard methodologies (according to international recommendations – e.g. EUCAST) prefer values in μg/mL, so we also measured MIC in this unit. But for calculations of SI, we have calculated also MIC value for Mtb in μM using molecular weight. Value of MIC is usually taken as 95-99% of inhibition. Different percentages would be highlighted as IC with lower index number corresponding to percentage of inhibition.

The reviewer’s intent was not to criticise the choice of calculating MIC values as either μg/mL or μM. However, when discussing the results, it would be better to use the same unit of measure (see page 1 lines 34, 37, 38). As for the specification of the MIC99, the reviewer agrees with the authors; the comment was purely meant as a suggestion, it is far from being a necessary modification.

According to the journal reference style, issue numbers are neither required nor used in recent Biomolecules' articles. Where available, the DOIs and pages were supplemented.

The authors are correct. The reference to the “issue number” was actually an oversight on the reviewer’s part. The indication was referred to the missing “volume” of reference 42 (Omosa et al.).

Thank you very much for your valuable feedback and time spent re-checking the manuscript.

In light of these changes, we are positive that our revised manuscript meets the criteria to be published in Biomolecules and would be of interest for all readers from the scientific community, with particular emphasis on drug development against mycobacterial infections.

Lucie Cahlikova